# INCORPORATING HUMAN PREFERENCES INTO INTERPRETABLE REINFORCEMENT LEARNING WITH TREE POLICIES

## ABSTRACT

Interpretable reinforcement learning (RL) seeks to create agents that are efficient, transparent, and understandable to the populations that they impact. A significant gap in current approaches is the underutilization of human feedback, which is typically employed only for post-hoc evaluation. We propose to center the needs of end users by incorporating the feedback that would be obtained in a user study directly into the training of interpretable RL algorithms. Our approach involves preference learning, where we learn preferences over high-level features that are not directly optimizable during the RL training process. We introduce an evolutionary algorithm that leverages user feedback to guide training toward interpretable decision-tree policies that are better aligned with human preferences. We demonstrate the effectiveness of our method through experiments using synthetic preference data. Our results show an improvement in preference alignment compared to baselines, yielding policies that are more aligned with underlying user preferences but does so with sample efficiency in the number of user queries, thereby decreasing the burden on the user in providing such data.

## 1 INTRODUCTION

Advancements in reinforcement learning (RL) have resulted in autonomous agents capable of performing tasks with remarkable ability in many applications, such as robotics (Mahmood et al., 2018), games (Vinyals et al., 2019; Hafner et al., 2023), vehicular control (Yan et al., 2022), and more. However, these achievements often come at the cost of interpretability: the neural networks governing the agents' decision-making processes are opaque, hindering understanding of their workings (Räuker et al., 2023). This opacity not only hinders people's ability to decide to trust and collaborate with these agents but also poses significant challenges in diagnosing and rectifying their faults.

Interpretable RL seeks to address this challenge by generating understandable representations of these decision-making processes (Glanois et al., 2021). However, existing techniques often do not account for user needs or preferences. From the perspective of human-computer interaction, adopting a user-centered approach is fundamental for ensuring that user needs are met (Lieberman et al., 2006). Recognizing this, researchers in the interpretable machine learning field have increasingly advocated for integrating human-centered methodologies into both the design (Schoonderwoerd et al., 2021) and evaluation (Boyd-Graber et al., 2022; Colin et al., 2022) of interpretable models.

In interpretable RL, however, the goal of satisfying end user desires is, in practice, often not reached. Evaluations of explanation quality and utility are frequently performed only after training (Narayanan et al., 2022), so insights garnered from these evaluations are not readily integrated back into the model itself. We envision utilizing human feedback during *training* to find RL policies that are better aligned with the preferences and desired tasks of users. In this work, we focus on constructing interpretable decision tree (DT) policies (McCallum, 1996; Pyeatt et al., 2001; Gupta et al., 2015) instead of opaque neural network policies. DTs (Quinlan, 1996) are a popular method for interpretable machine learning due to the consensus that they are human-understandable (Lipton, 2018; Rudin, 2019).

We propose to center user preferences through a novel learning approach that incorporates human feedback into the training process of interpretable RL algorithms. The algorithm iteratively refines its estimate of the underlying user preferences over DT policies and constructs new candidate policies to

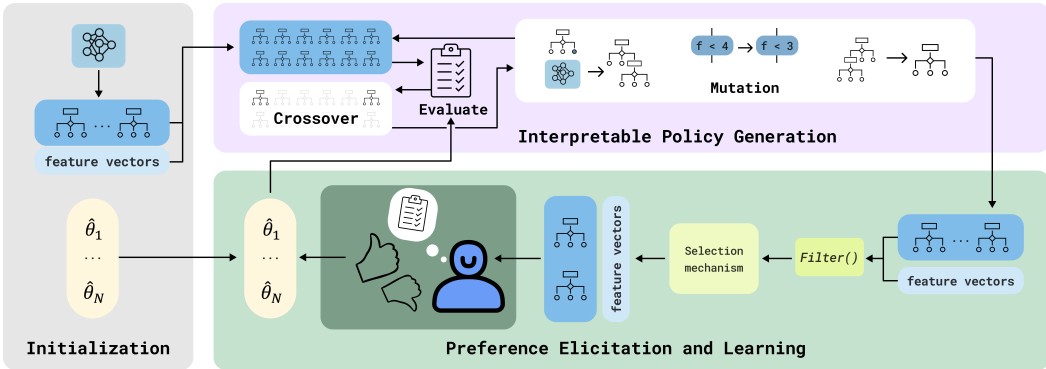

Figure 1: **PASTEL overview.** PASTEL consists of two main parts: i) preference elicitation and learning and ii) interpretable policy generation. In preference elicitation and learning, users provide feedback on candidate interpretable models. During interpretable policy generation, the current preference estimate guides generation of interpretable models that better align with user preferences.

query for user preference feedback. The key insight is that maintaining a set of preference estimates can be useful for constructing DT policies that maximally satisfy those estimates. Then, eliciting further feedback by potentially considering these candidate policies allows us to refine the estimate. By incorporating this feedback during training, we create policies that are better aligned with user needs and preferences, rather than simply evaluating or measuring these preferences after training.

**Contributions** In this work, we make the following contributions. We propose "Preference Aligned Selection of Trees via Evolutionary Learning" (PASTEL), the first algorithm to leverage preference feedback for interpretable RL training. This algorithm consists of three main components. First, we introduce a simple but effective evolutionary algorithm to guide training toward DT policies that are better tailored toward preferences of end users. Second, we maintain a population of preference estimates to help guide the selection of DT policies used for eliciting user feedback. Third, we improve efficiency by pruning the set of candidate DT policies for querying, maintaining an approximation of the Pareto frontier. We demonstrate the effectiveness of our method through experiments on two RL environments using synthetic but empirically justified preference data. We show that our approach not only yields policies that are more aligned with user preferences but is also more sample efficient in the number of user queries, decreasing the burden on human users. By bridging the gap between training RL agents and evaluating their explanations, we believe our work opens new avenues for developing more interpretable, user-centered RL systems. In summary, we contribute the following:

- A framework formalizing the idea of preference-based interpretable reinforcement learning,
- PASTEL, the first algorithm to leverage preference feedback to train interpretable RL policies, and
- An evaluation of PASTEL showing that it can produce better preference-aligned DT policies compared with standard techniques for generating interpretable DT policies.

## 2 RELATED WORK

**Interpretable Machine Learning.** Interpretable machine learning (ML) has become increasingly important as ML systems are deployed for more complex and sensitive use cases (Prashanth et al., 2016; Chebotar et al., 2021). The need for interpretability arises from the desire to understand, appropriately trust, and effectively manage these systems. The most relevant line of work to ours is in interpretable RL. Although many structures have been proposed to use in lieu of neural networks for RL policies, including domain-specific programming languages (Verma et al., 2018) and human-friendly prototypes (Kenny et al., 2022), we focus on DT policies due to the general consensus that they are human-understandable (Lipton, 2018; Rudin, 2019). One challenge with DTs is their discrete structure is not immediately amenable to gradient-based training. Although work has applied gradient-based training to DT policies by introducing non-linearities in the splits (Silva et al., 2020),

the conversion of the soft tree (Irsoy et al., 2012) to a standard DT results in severe performance loss, making this approach inapplicable in our setting. Moreover, prior work trains DT policies only from the perspective of reward maximization (Bastani et al., 2018; Topin et al., 2021) under a fixed interpretability objective (typically, depth minimization). Our approach not only focuses on obtaining reward-maximizing DT policies but also aligning them with preferences. While our emphasis is on interpretability, we note that our framework also readily lends itself to explanation generation.

**Learning from Human Feedback.** There is growing interest in the area of learning from human feedback (Griffith et al., 2013; Hadfield-Menell et al., 2016; Stiennon et al., 2020) due to the difficulty of specifying a concrete objective that captures the full extent of what people want ML algorithms to do. Our work can be viewed as a general framework for incorporating human feedback into interpretable machine learning. In contrast, previous work often treats the interpretability objective as fixed (e.g., by specifying a maximum tree depth or known weight on the depth (Custode and Iacca, 2023)) or only as a way to *evaluate* the resulting explanations. This prior knowledge can be used to set an initial guess within our model, and the evaluation metrics can be used along with our approach. Furthermore, we aim for a practical algorithm that demonstrates sample efficiency in gathering preferences, aligning with prior work that emphasizes the importance of minimizing user effort in interactive machine learning systems (Cakmak and Thomaz, 2012; Wilson et al., 2012).

**Preference Elicitation.** Preference elicitation and learning have been extensively studied across various disciplines, addressing the need to understand preferences over a set of alternatives (Chajewska et al., 2000; Conen and Sandholm, 2001; Loepp et al., 2014; Weernink et al., 2014; Li et al., 2023). Approaches in this field often focus on ranking elicitation by employing methods such as pairwise comparisons (Eric et al., 2007; Lu and Boutilier, 2011a; Branke et al., 2017) or asking individuals to rank their top choices among all or a subset of alternatives (Lu and Boutilier, 2011b; Soufiani et al., 2013; Drummond and Boutilier, 2014; Zhao et al., 2018). These methods typically result in the elicitation of full rankings or partial orders for agents or groups. Our work, while building on these works on preference elicitation, shifts the focus from eliciting rankings to learning the underlying preference model parameters. Our approach shares similarities with Bayesian Experimental Design (Chaloner and Verdinelli, 1995; Rainforth et al., 2024), where the objective is to learn information about the preference model $\theta$, and aims to explore the integration of preference elicitation and learning, with an emphasis on aligning models with the preferences of downstream users.

## 3 PROBLEM STATEMENT

We formulate the problem of integrating user feedback into interpretable RL to identify the most preferred DT policy. We frame the task as an online iterative process of pairwise comparisons over a finite horizon, or query budget, $T$.

**Interpretable Tree Policy Representation.** Let $\hat{\Pi}$ denote the set of all possible DT policies and $\mathcal{E}$ represent the set of all possible environmental or contextual information. We define a mapping function $m : \hat{\Pi} \times \mathcal{E} \to \mathbb{R}^d$ that maps a DT policy $\hat{\pi}$ and the full set of environmental information to a $d$-dimensional feature vector $\mathbf{f}_{\hat{\pi}} \in \mathbb{R}^d$, such that

$$\mathbf{f}_{\hat{\pi}} = m(\hat{\pi}, \mathcal{E}). \tag{1}$$

These features may include structural attributes (e.g., depth, number of leaves), performance metrics (e.g., average accumulated reward), and explanation characteristics (e.g., which state features are used in the tree). We assume that the mapping $m$ is known and is sufficient to describe the users' preferences. The mapping function $m$ internally determines which elements are relevant and how they contribute to the resulting feature vector $\mathbf{f}_{\hat{\pi}}$.

**User Preference Model.** Let $\theta$ be a latent parameter vector representing user preferences. Then, following standard assumptions of linearity in the preference elicitation literature (Li et al., 2010; Chu et al., 2011; Saha, 2021), we model the user's utility function as a linear combination of the policy attributes and $\theta$, such that

$$v(\hat{\pi}; \theta) = v(m(\hat{\pi}, \mathcal{E}); \theta) = \mathbf{f}_{\hat{\pi}}^{\top} \theta. \tag{2}$$

This formulation allows us to compactly capture user preferences over various aspects of the DT policies. The user's stochastic response $o$ to pairwise queries of DT policies $Q = (\hat{\pi}, \hat{\pi}')$ is based on

their underlying utility function $v$. Given two DT policies $\hat{\pi}_i, \hat{\pi}_j$ with corresponding feature vectors $\mathbf{f}_{\hat{\pi}_i}, \mathbf{f}_{\hat{\pi}_j}$, the probability that $\hat{\pi}_i$ is preferred to $\hat{\pi}_j$ is:

$$\mathbb{P}(\hat{\pi}^i \succ \hat{\pi}^j | \theta) = \frac{1}{1 + \exp\left(\beta\Big(v\left(\mathbf{f}_{\hat{\pi}_j}; \theta\right) - v\left(\mathbf{f}_{\hat{\pi}_i}; \theta\right)\Big)\right)}, \tag{3}$$

where $\beta$ is an inverse temperature parameter governing the stochasticity of user responses. This preference model corresponds to the standard Terry-Plackett-Luce model (Bradley and Terry, 1952; Luce, 2012) with a rationality coefficient (Shah et al., 2016; Laidlaw and Dragan, 2022).

**Preference Elicitation Framework.** At each iteration $t \in [T]$, the algorithm can present the user with a pair of tree policies $Q_t = (\hat{\pi}_t, \hat{\pi}'_t)$ selected from a set $\hat{\Pi}_t$. The user provides stochastic feedback $o_t$ based on their underlying utility function $v$. We focus on the *pairwise* query setting, such that each query $|\mathcal{Q}_t| = 2$.

**Learning Objective.** After $T$ queries, the learner outputs a recommended tree $\hat{\pi}_T$. The learner derives the recommendation as:

$$\hat{\pi}_T = \operatorname{argmax}_{\hat{\pi} \in \hat{\Pi}} \hat{v}_t(\hat{\pi}; \hat{\theta}), \tag{4}$$

where $\hat{v}_T(\hat{\pi}; \hat{\theta})$ is the estimated utility of tree $\hat{\pi}$ after $T$ queries, and $\hat{\Pi}_T$ is the set of trees considered up to iteration $T$. We formulate the objective as maximizing the true utility of the recommended DT policy after $T$ queries:

$$\max_{\hat{\pi} \in \hat{\Pi}} v(\hat{\pi}; \theta). \tag{5}$$

A key advantage of our setting is the ability to generate new interpretable models or explanations, providing a higher degree of control over the options presented to users in each round. This contrasts with settings like movie recommendation, where creating new items based on preferences is impractical.

## 4 PASTEL

How can we create interpretable policies that align with what humans actually want? We propose PASTEL, an algorithm that interleaves preference learning and policy finding to produce interpretable decision tree policies aligned with user preferences. Detailed in Algorithm 1, this algorithm proceeds as follows. First, PASTEL creates initial preference estimates $\hat{\theta} = \hat{\theta}_i, \dots \hat{\theta}_N$ and an initial population of DT policies $\hat{\Pi}_0$. In each iteration $t \in [T]$, the algorithm first updates preference estimates and then updates the policy population by repeating steps 2 and 3:

1. **Initialization**: Create a set of preference estimates and an initial population of DT policies.
2. **Preference Elicitation and Learning**: Select policies for querying the user, obtain user feedback, and update the preference estimate $\hat{\theta}_t$.
3. **Interpretable Policy Generation**: Use an evolutionary algorithm to generate a new population of policies $\hat{\Pi}_t$.

After exhausting the query budget, the algorithm identifies the best policy with respect to the final preference estimate. We now describe each component in detail.

### 4.1 INITIALIZATION

**Create Initial Preference Estimates.** We initialize multiple preference estimates $\{\theta_1 \dots \theta_N\}$ using Xavier initialization, where each $\theta_i \in \mathbb{R}^d$ is drawn from a uniform distribution $\mathcal{U}(-r, r)$ with $r = \sqrt{6/d}$. This approach, inspired by the Query by Committee (QbC) paradigm in active learning, maintains a diverse set of hypotheses about user preferences. The ensemble of estimates $\Theta = \theta_1, \dots, \theta_N$ enables robust exploration of the preference space $\mathcal{P} \subseteq \mathbb{R}^d$, mitigating the risk of overfitting to early, potentially noisy preferences. For policy selection, we employ a voting mechanism where each estimate $\theta_i$ contributes to the decision. We simply take the maximum, breaking ties uniformly at random.

---

**Algorithm 1** PASTEL

---

1: Initialize $\hat{\theta} = \hat{\theta}_0^1, \ldots, \hat{\theta}_0^N$ randomly
2: Train RL policy $\pi^* \leftarrow$ RLTraining()
3: Generate initial population of tree policies $\hat{\Pi}_0 = \{\hat{\pi}_1, \ldots \hat{\pi}_{K \times M}\}$ using VIPER-style imitation learning of $\pi^*$
4: Obtain feature vector $\mathbf{f}_{\hat{\pi}}$ for each $\hat{\pi} \in \hat{\Pi}$
5: Initialize best policy so far as $\hat{\pi}_0^* \leftarrow \hat{\pi} = \arg\max_{\hat{\pi} \in \hat{\Pi}_0} \text{vote}(\{\hat{v}_i(\mathbf{f}_{\hat{\pi}}; \theta_i)\}_{i=1}^N)$
6: **for** $t \leftarrow 1$ **to** $T$ **do**
7:     Obtain reduced population $\hat{\Pi}_{t-1}$ with Pareto frontier filtering
8:     Choose policies to present to user according to Equation (7)
9:     Obtain feedback $o_{t-1}$ about $Q_{t-1}$ according to Equation (3)
10:     Update $\hat{\theta}_t^i$ according to Equation (8)
11:     **for** $\hat{\theta}_t^i$ **do**
12:         Assign the initial population for EA $\Psi_0 \leftarrow \hat{\Pi}_0$
13:         **for** $g = 0, 1, \ldots, G-1$ **do**
14:             Evaluate fitness of $\Psi_g$ according to $\mathbf{f}_{\hat{\pi}}^\top \hat{\theta}_t^i$
15:             Initialize next population $\Psi_{g+1} \leftarrow \emptyset$
16:             **while** $|\Psi_{g+1}| <$ population size **do**
17:                 $p_1, p_2 \leftarrow$ SelectParents($\Psi_g, k$) using tournament selection
18:                 **if** $n \sim \mathcal{U}(0,1) < \rho$ **then**
19:                     $p_1, p_2 \leftarrow$ Crossover($p_1, p_2$)
20:                 **end if**
21:                 Mutate($p_1$), Mutate($p_2$)
22:                 Add $p_1, p_2$ to $\Psi_{g+1}$
23:             **end while**
24:         **end for**
25:     **end for**
26:     Assign $\hat{\Pi}_t \leftarrow \Psi_G^i, P$ for more preference queries
27: **end for**
28: **return** Best policy $\hat{\pi}^*$ according to $\hat{v}_T$

---

**Generate Policies for Preference Elicitation.** In real-world applications, policies typically must not only satisfy user preferences but also perform competently in their intended domain. We leverage this insight to develop an initialization method for candidate policies that are both performant and diverse with the goal of providing a strong starting point for preference elicitation. Our approach builds on VIPER (Bastani et al., 2018), a DAgger-based interactive imitation learning algorithm (Ross et al., 2011) in which an RL policy $\pi^*$ acts as an expert to guide the training of decision-tree policies. Following this framework, we train a high-quality RL policy $\pi^*$ for the domain of interest. Then, we generate a pool of DTs, which serve as candidate tree policies. To ensure that there is sufficient diversity in the initial set of DT policies, we make two key modifications to VIPER. First, instead of only outputting the final best tree $\hat{\pi}$ according to return or alignment with the expert $\pi^*$, we instead use *all* tree policies generated during the $K$ VIPER iterations. Second, we introduce *depth randomization*, in which we choose a set of $M$ maximum depths to constrain the resulting depths of the trees. From this process, we obtain an initial population of $|\hat{\Pi}_0| = K \times M$ tree policies, which we then convert into their corresponding $d$-dimensional feature vectors $\mathbf{f}_{\hat{\pi}} \in \mathbb{R}^d \leftarrow m(\hat{\pi}, \cdot), \forall \hat{\pi} \in \hat{\Pi}_0$.

**Pareto Frontier Filtering.** Given that users have limited time, we want to ensure that the feedback we gain from each comparison is informative. To maximize the value of each user interaction, we avoid presenting policies that are dominated in all dimensions, as repeated selections of clearly superior policies provide little information about how users trade off different feature dimensions. Instead, we propose filtering the policy set based on the Pareto frontier. Given a dataset $D$ in a multi-dimensional space and a corresponding preference direction for each dimension, we identify the points that comprise the Pareto frontier:

$$\mathcal{P} = \{x \in D \mid \nexists y \in D, \text{ such that } y \text{ dominates } x\}, \tag{6}$$

where a point $y$ dominates $x$ if it is at least as good in all dimensions and strictly better in at least one dimension. PASTEL then chooses from this reduced set of trees for preference elicitation. This

approach not only ensures that users are presented with only the most promising options but can also dramatically reduce the computational complexity of the selection process. In many practical cases, the size of the Pareto frontier grows sublinearly with respect to $n$. For instance, in two-dimensional problems, it follows a logarithmic relationship. This reduces the complexity from the naive $O(Tn^2)$ to $O(T\log^2 n)$. This reduction means that we can explore the use of more computationally intensive techniques for identifying the most informative pairs for comparison.

## 4.2 PREFERENCE ELICITATION AND LEARNING

We now introduce our preference elicitation and learning process, designed to learn user preferences over DT policies. Our approach combines a selection strategy with an update mechanism to navigate preference spaces. We aim to converge on accurate preference estimates while handling stochasticity in user responses.

**Selection Process** The selection process employs a novel approach to identify the most informative pair of items $(\mathbf{f}_{\hat{\pi}_i}, \mathbf{f}_{\hat{\pi}_k})$ from a set $F$, given a collection of preference estimates $\theta_j j = 1^m$. For each potential pair and each $\theta_j$, the algorithm computes two hypothetical updates:

$$\theta_j^{(i)} = \theta_j - \eta\nabla\mathcal{L}(\mathbf{f}_{\hat{\pi}_i} \succ \mathbf{f}_{\hat{\pi}_k}|\theta_j) \text{ and } \theta_j^{(k)} = \theta_j - \eta\nabla\mathcal{L}(\mathbf{f}_{\hat{\pi}_k} \succ \mathbf{f}_{\hat{\pi}_i}|\theta_j),$$

where $\eta$ is the learning rate and $\nabla\mathcal{L}$ is the gradient of the logistic loss. The cosine similarity $\cos(\theta_j^{(i)}, \theta_j^{(k)})$ is then computed for each $\theta_j$. The algorithm selects the pair $(\mathbf{f}_{\hat{\pi}_i}, \mathbf{f}_{\hat{\pi}_k})$ that minimizes the average cosine similarity across all $\theta_j$:

$$(\mathbf{f}_{\hat{\pi}_i}, \mathbf{f}_{\hat{\pi}_k}) = \arg\min_{(\mathbf{f}_{\hat{\pi}_i}, \mathbf{f}_{\hat{\pi}_k})} \frac{1}{m} \sum_{j=1}^{m} \cos(\theta_j^{(i)}, \theta_j^{(k)}). \tag{7}$$

This approach aims to maximize the expected information gain by choosing pairs that lead to the most orthogonal (i.e., least similar) updates across the ensemble of preference estimates, thereby efficiently exploring the preference space.

**Preference Update** In modeling preferences over DT policies, we seek an approach that accounts for uncertainty, allows for online updates, and efficiently estimates preferences in a potentially high-dimensional feature space. As a result, we adopt a logistic regression model with stochastic gradient updates. After receiving (noisy) feedback $o_t$ about $Q_t$ from the user based on Equation (3), the algorithm then updates the current estimates of the underlying preference vector. Specifically, PASTEL uses the user feedback and the selected DT policies' corresponding feature vectors $\mathbf{f}_{\hat{\pi}}, \mathbf{f}'_{\hat{\pi}}$ to update each $\hat{\theta}$ as:

$$\hat{\theta}_{t+1} = \hat{\theta}_t + \alpha \cdot (o_t - \sigma(\hat{\theta}_t^\top(\mathbf{f}_{\hat{\pi}} - \mathbf{f}'_{\hat{\pi}}))) \cdot (\mathbf{f}_{\hat{\pi}} - \mathbf{f}'_{\hat{\pi}}), \tag{8}$$

where $\alpha$ is the learning rate, $\sigma(\cdot)$ is the logistic function, $o_t \in \{0, 1\}$ is the binary label derived from the user's feedback, and $\mathbf{f}_{\hat{\pi}} - \mathbf{f}'_{\hat{\pi}}$ is the difference between the feature vectors of the compared DT policies.

## 4.3 INTERPRETABLE POLICY GENERATION

Equipped with our updated $N$ preference estimates, we now focus on generating new structures to align with those preferences. The goal is to create these structures such that they can be checked against (and possibly added to) the current Pareto frontier for further preference elicitation. Because many popular structures used in interpretable reinforcement learning, such as DTs, are not differentiable, we require an algorithm that can produce new DTs without relying on gradients.[1] As a result, we introduce an EA to generate new DT policies. EAs (Bäck and Schwefel, 1993) do not require the objective function to be differentiable. They work by evolving a population of solutions over generations to optimize a given objective function, called the fitness function.

In PASTEL, we call an EA for each of the theta estimates $\hat{\theta}_i, \ldots, \hat{\theta}_N$, such that the fitness function for the associated EA uses $\hat{\theta}_i$ to evaluate the quality of each individual in the population. Specifically,

---

[1]When the interpretable models or explanations can be updated using gradients, one can directly leverage work that uses preference feedback for RL (Christiano et al., 2017; Bai et al., 2022).

each $\mathbf{f}_{\hat{\pi}}$ is evaluated as $\hat{v}(\mathbf{f}_{\hat{\pi}}) = \hat{\theta}\mathbf{f}_{\hat{\pi}}$, and the individual that maximizes this value after $G$ rounds of the EA is the best tree policy according to that $\hat{\theta}$ estimate. Then, it is compared with the set of DT policies in the Pareto frontier and added to the set if it is non-dominated. Each time the EA is called it proceeds as in Lines 17-29 of Algorithm 1.

**Fitness** Because we envision that this algorithm could actually interact with end users, we prioritize efficiency. One challenge is that, in each EA loop, we need to estimate the quality, or fitness, of each new DT policy using $\hat{\theta}$. When the feature vector comprises of simple policy attributes, this estimation is fast. However, when environment return is included, we often need to perform rollouts for a fixed number of episodes $\rho$ to estimate the expected return. This process could be computationally expensive, especially with a large number of candidates. To address this challenge, we propose an adaptive policy evaluation algorithm that dynamically adjusts its evaluation process based on the policy's performance profile, allowing for fewer episodes to be evaluated for less promising candidates. We use an UCB-inspired bound (Slivkins et al., 2019) to estimate the expected return for each tree as

$$\tilde{R} = \hat{R} + c\sqrt{\frac{\ln(n)}{n}}, \tag{9}$$

where $n$ is the number of episodes performed. As we gather more samples (increasing $n$), our confidence interval narrows, allowing us to be more certain estimate on the expected return of each arm. To allow for fewer episodes to be rolled out for less promising arms, we design a stopping condition

$$c\sqrt{\frac{\ln(n)}{n}} \leq \frac{\epsilon_{\text{base}}}{1 + \epsilon_{\text{scale}} \cdot \tilde{R}}, \tag{10}$$

which is updated after each rollout. This formulation embodies a key insight: as the estimate $\tilde{R}$ increases (suggesting a potentially better policy), we decrease $\epsilon$, demanding higher precision. This allows us to differentiate more accurately between high-performing policies.

**Crossover** In EAs, the crossover operator aims to construct new individuals by combining the parts of promising individuals. We choose to preserve individuals that are scored highly according to $\hat{\theta}$, then perform subtree swapping as a crossover operator. Subtree swapping first randomly selects a node within the tree structure of the two parents in the crossover operation, denoted as $n_{\text{cross}} \in N$. Then, the subtrees rooted at this node are exchanged, resulting in two new individuals.

**Mutation** To more broadly explore the space of possible trees, we perform mutations on randomly selected nodes of a tree in one of three ways, determined by tunable weights on the three mutation strategies: random, subtree rebuilding, and subtree removal. Random mutation ($\mathcal{M}_{\text{rand}}$) selects a random feature and a corresponding split value (or selects a new random action) to create a new node $n'$. Subtree rebuilding ($\mathcal{M}_{\text{subtree}}$) first retrieves the subset of data $\mathcal{D}_{\text{path}} \in \mathcal{D}$ corresponding to a chosen node. The subtree is then reconstructed using imitation learning with $\mathcal{D}_{\text{path}}$. Subtree removal replaces the subtree rooted at the selected node with an action.

## 5 EXPERIMENTS

We evaluate the performance of PASTEL on two different RL environments. Because DTs are generally considered interpretable, we conduct a *functionally-grounded* evaluation (Doshi-Velez and Kim, 2017), in which we define a specific set of features that would inform user preferences for interpretable models and simulate these preferences. A crucial advantage of testing our method in these simulated environments is that we can evaluate how well our model is able to recover the preferences of the "ground truth" users, which provides us with insights about how our methods could perform with real users.

**Environments.** We select CartPole-v1 and PotholeWorld-v0 as our test environments not only because a reward-optimal policy can be represented as a tree but also for their complementary characteristics. CartPole-v1, a classic control problem, offers a simple, well-understood domain with a low-dimensional state space (cart position, velocity, pole angle, and angular velocity) and binary actions (push left or right), serving as an excellent benchmark for basic RL capabilities. In contrast, PotholeWorld-v0 (Topin et al., 2021) presents a more complex, driving-inspired scenario

| CartPole-v1 | | | PotholeWorld | | |
|---|---|---|---|---|---|
| Feature | Type | Values | Feature | Type | Values |
| Reward | Performance | $[0, 500]$ | Reward | Performance | $[-170, 49.95]$ |
| Depth | Structural | $\{0, \ldots, 10\}$ | Depth | Structural | $\{0, \ldots, 10\}$ |
| Num. leaves | Structural | $\{1, \ldots, 1024\}$ | Num. leaves | Structural | $\{1, \ldots, 1024\}$ |
| State feature used | Explanation | $\{0, 1\}$ | Action taken | Explanation | $\{0, 1\}$ |

Table 1: Features used in the preference vectors for the two environments.

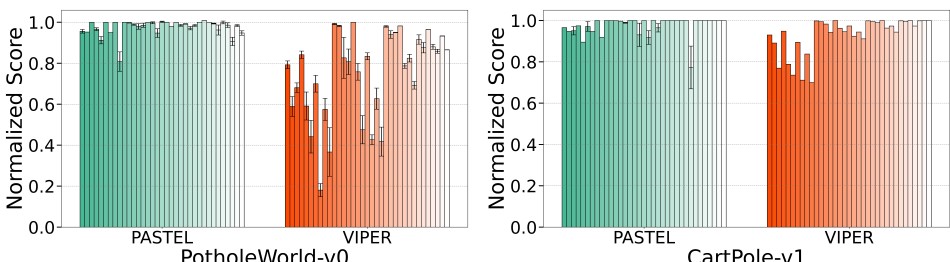

Figure 2: Comparison of normalized scores for PASTEL and VIPER algorithms on the PotholeWorld-v0 and CartPole-v1 environments. Each bar in the plots represents a different preference vector.

where an agent navigates lanes to avoid obstacles. We augment PotholeWorld-v0 with a controllable parameter governing lane reward trade-offs, allowing exploration of long-term planning in stochastic environments. In PotholeWorld-v0, the state is simply the current position, with three possible actions corresponding to lane choices. As a result, the agent must learn through experience to navigate around the potholes.

**Features.** We test with $d = 4$ features for each environment. To evaluate the method, we choose at least one feature from each type (performance-based, structural, and explanation characteristics). Table 1 shows the features used.

**Preference Generation.** To generate the values for $\theta$, we adopt a vector scaling technique (Ehrgott, 2005). We linearly combine the basis vectors, with each vector maximizing only one preference. This setup enables the exploration of explicit trade-offs among these preferences by adjusting the scaling factor for each vector entry $\alpha$, such that $\sum_{i=1}^{n} |\theta_i| = 1$. The scaling factor is 0.25, resulting in 35 different $\theta$ values for each environment. We run each $\theta$ with 3 random seeds for each experiment.

**Baselines** In our experiments, we compare against two baselines: VIPER and what we call Randomized Dueling Preference Selection (RDPS). The latter baseline is inspired by randomized dueling bandit algorithms used in preference-based learning. The algorithm maintains the current best explanation. At each iteration $t$, the algorithm compares the current best with a randomly selected challenger. After eliciting pairwise preference feedback, the winner of the duel becomes the updated best explanation. This process is repeated for $T$ queries.

**Performance Metrics.** We normalize the results using the maximum and minimum possible values of the decision tree, obtained by running EA for each true preference vector, $\theta$, and its inverse, $-\theta$, respectively. The normalized values are in $[0, 1]$.

### 5.1 RESULTS

**PASTEL produces tree policies that better align with human preferences compared with VIPER.** We first investigate whether PASTEL produces better-aligned DT policies under a noisy regime by setting $\beta$ in Equation (3) to be 10. We compare PASTEL with VIPER to verify that we can indeed produce better preference-aligned trees than VIPER's reward maximizing algorithm. Because VIPER does not incorporate preferences in its learning process, the best tree is chosen based on return only. We evaluate the quality of the DT policies recommended by both approaches at $T = 40$ queries for PotholeWorld-v0 and $T = 20$ queries for CartPole-v1. As shown in Figure 2, PASTEL produces substantially more preference-aligned trees than VIPER. For PotholeWorld-v0, PASTEL

| Noise Regime | PASTEL | RDPS | VIPER |
|---|---|---|---|
| Easy | 0.9772 ± 0.0042 | 0.89649 ± 0.0114 | 0.75628 ± 0.0328 |
| Medium | 0.9496 ± 0.0089 | 0.843133 ± 0.0160 | 0.75628 ± 0.0328 |

Table 2: Mean normalized scores for PASTEL, RDPS, and VIPER in two different noise regimes.

| Algorithm | PotholeWorld-v0 | CartPole-v1 |
|---|---|---|
| PASTEL | 0.9772 ± 0.0042 | 0.9865 ± 0.0028 |
| PASTEL-DTEA | 0.9349 ± 0.0043 | 0.8872 ± 0.0204 |
| PASTEL-ITER | 0.7158 ± 0.0470 | 0.9864 ± 0.0030 |
| PASTEL-PFF | 0.9668 ± 0.0054 | 0.9642 ± 0.0081 |
| PASTEL-CSS | 0.9707 ± 0.0052 | 0.9884 ± 0.0027 |

Table 3: Ablation study: mean normalized scores for all PASTEL ablations. Error is standard error. For PotholeWorld-v0, all components of PASTEL contribute to its performance. However, CartPole-v1 is simpler, so the major benefit stems from the EA.

maintains scores close to 1.0 with minimal variance, indicating robust and near-optimal performance. In contrast, VIPER shows considerable fluctuation in its scores. Similarly, in CartPole-v1, PASTEL again demonstrates higher stability and performance, consistently achieving near maximum scores across all preference vectors. VIPER's performance in CartPole-v1, while showing some improvement over its PotholeWorld-v0 results, still exhibits significant variability with scores ranging from around 0.6 to 1.0. These findings suggest that PASTEL can indeed produce more preference-aligned DT policies.

**PASTEL produces aligned policies under different levels of noise.** We investigate this question in PotholeWorld-v0 for two different noise regimes: easy and medium. We achieve these regimes by setting $\beta$ in Equation (3) to values that result in progressively *more* probabilistic disparity between the best and the worst trees in the original set of trees. In other words, there exists a $1 - \epsilon$ probability of choosing the best tree over the worst tree. In the medium setting, we set $\beta$ such that $\epsilon$ is closer to 0.5 and reduce the value to reflect the difficulty of the other settings. Here, we compare with RDPS and VIPER. However, VIPER does not learn preferences and therefore is resistant to noise. Table 2 shows the results of this experiment. We find that, although RDPS can outperform VIPER in terms of preference score, PASTEL still can find better preference-aligned trees in both noise regimes. This indicates that PASTEL can be relatively robust to noise.

**Ablation study: the different components of PASTEL have different, environment-dependent contributions to the overall score.** Table 3 presents the results of an ablation study on the PASTEL algorithm. The base version of PASTEL achieves strong performance in both environments, with mean normalized scores of 0.9772 ± 0.0042 and 0.9865 ± 0.0028, respectively. Removing the evolutionary algorithm component (PASTEL-DTEA) results in a notable drop in performance, especially in CartPole-v1, where the score decreases to 0.8872 ± 0.0204, suggesting that the evolutionary strategy is particularly important in environments with lower complexity. Similarly, the non-iterative variant (PASTEL-ITER) shows a significant degradation in PotholeWorld-v0, with a mean score of 0.7158 ± 0.0470, while maintaining comparable performance in CartPole-v1 (0.9864 ± 0.0030). The absence of Pareto frontier filtering (PASTEL-PFF) leads to a slight reduction in both environments, indicating that Pareto filtering contributes modestly but consistently to overall performance. Lastly, random sampling (PASTEL-CSS) instead of the designed preference elicitation strategy leads to only a minor performance drop, with the algorithm still maintaining competitive scores in both environments. These results show that each component of PASTEL contributes to its performance, though the importance of each varies depending on the environment.

# 6 CONCLUSION AND FUTURE WORK

We proposed to incorporate user feedback directly into the explanation-generation process. This approach provides a proof of concept that incorporating the preferences of users into the process yields explanations that are better aligned with users goals and desires. By leveraging human feedback during training, our approach addresses a critical gap in existing interpretable RL methods,

which often fail to consider user needs in real-time. Through experiments in two environments, we demonstrated that our method not only produces DT policies that are better aligned with the underlying preferences but also does so efficiently, in 40 queries or fewer. Our work lays the groundwork for future research on developing more user-centered, interpretable RL systems that prioritize alignment with human expectations.

**Limitations and Future Work.** One limitation of our work is the assumption of static user preferences, which may not accurately reflect the dynamic nature of preferences in real-world applications. To address this, future work could explore techniques for modeling and adapting to evolving user preferences over time, allowing the system to update and refine policies as user goals shift. Another limitation is the use of synthetic data for evaluation, which may not fully capture the diversity of real-world user perspectives. In future studies, incorporating a wider range of user input and adopting an *application-grounded* evaluation approach (Doshi-Velez and Kim, 2017) could provide more meaningful insights, particularly by leveraging real-world data to model the latent factors that influence user preferences. This would enable more robust, user-centered policy generation in practical settings.

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
