Table 4: Hyperparameters of the evolutionary algorithm for generating the upper and lower bounds.

| Hyperparameter | PotholeWorld-v0 | CartPole-v1 |
|---|---|---|
| Population Size | 75 | 75 |
| Mutation Rate $\mu$ | 0.9 | 0.9 |
| Crossover Rate $\rho$ | 0.8 | 0.8 |
| Tournament Size $k$ | 3 | 3 |
| Max Generations | 125 | 100 |

Table 5: VIPER algorithm hyperparameters

| Hyperparameter | PotholeWorld-v0 | CartPole-v1 |
|---|---|---|
| Expert Algorithm | DQN | DQN |
| Number of Batch Rollouts | 10 | 10 |
| Maximum Samples | 200000 | 200000 |
| Maximum Iterations | 20 | 20 |
| Training Fraction | 0.8 | 0.8 |
| Reweighting | True | True |
| Number of Test Rollouts | 50 | 50 |

## A ADDITIONAL EA DETAILS

**Individual Encoding of Trees**    Our EA does not operate on the decision trees directly; instead, we represent each tree as a real-valued feature vector in continuous space. Each tree is vectorized in depth-first order, with all vectors having a consistent length determined by a pre-specified maximum depth $M$. Specifically, each feature vector has a dimension of $\mathbb{R}^{2^{M+1}-1}$, encoding the structure and information of the tree. This allows for more efficient manipulation and comparison within the continuous feature space, avoiding the need to store and operate on the full decision trees directly.

**Tournament Selection Mechanism**    For selecting individuals during reproduction, we use tournament selection with a size of 3, where three random individuals from the population compete based on their fitness, and the best among them is chosen as a parent. This process ensures that better-performing individuals are more likely to be selected while maintaining diversity by allowing weaker individuals a chance to reproduce.

## B HYPERPARAMETERS

In this section, we provide detailed descriptions of the hyperparameter settings used in our experiments.

**Evolutionary Algorithm for Upper and Lower Bounds**    We use the EA to arrive at the upper and lower bounds on the preference score for each preference for each environment. We present the specific configurations for the hyperparameters in Table 4. This table includes essential parameters such as the mutation rate, crossover rate, population size, and the number of generations, among others.

**VIPER**    The hyperparameters specific to the VIPER algorithm are detailed in Table 5, including the number of batch rollouts, maximum samples, and the fraction of data used for training. These number of batch rollout controls how many expert trajectories are generated during training, providing the data needed to distill the policy. The maximum number of samples limits the total state-action pairs collected. The fraction of data used for training determines how much of the collected dataset is used in each iteration, balancing the need for immediate policy updates with the potential benefit of retaining data for future use. The maximum iterations specifies how many rounds of VIPER are completed. Reweighting means that the algorithm reweights the samples used to train the DT according to the associated Q-values.

Table 6: Hyperparameters of PASTEL.

| Hyperparameter | PotholeWorld-v0 | CartPole-v1 |
|---|---|---|
| Population Size | 75 | 75 |
| Number of Trees in Initial Population | 675 | 150 |
| Mutation Rate $\mu$ | 0.9 | 0.9 |
| Crossover Rate $\rho$ | 0.8 | 0.8 |
| Tournament Size $k$ | 3 | 3 |
| Max Generations | 75 | 75 |
| Expert for Imitation Learning | DQN | DQN |
| EA Start (Query Number) | 5 | 5 |
| EA Spacing (every $k$ steps) | 5 | 5 |

**PASTEL** In Table 6, we provide the hyperparameters for PASTEL. Both environments, PotholeWorld-v0 and CartPole-v1, share similar configurations, with the only difference that PotholeWorld-v0 uses more trees in the initial (VIPER-generated) population.