# OpenReview forum: "Incorporating Human Preferences into Interpretable Reinforcement Learning with Tree Policies"
_ICLR.cc/2025/Conference — Submitted to ICLR 2025_

### Official Review · Reviewer_kmwo · 2024-10-31

**Soundness:** 3
**Presentation:** 2
**Contribution:** 2
**Rating:** 5
**Confidence:** 3

**Summary:**

Motivated by interpretability in reinforcement learning, this work proposes a method (PASTEL) for injecting human preferences into decision-tree policies. The general method uses an evolutionary strategy guided by user feedback, iteratively refining its estimate of user preferences and generating new candidate policies. Overall, experiments on CartPole and PotholeWorld demonstrate that PASTEL produces policies that are more aligned with user preferences, and are also more sample efficient compared to baselines like VIPER.

**Strengths:**

The paper clearly defines the problem, user preference model, and learning objective.
The paper has a well designed and modular algorithmic framework (though it is quite complex). Particularly: (1) the interpretable policy generation uses an evolutionary algorithm tailored to user preferences, generating new candidate policies that progressively align with user desires; and (2) PASTEL's Pareto frontier filtering and adaptive policy evaluation mechanisms aim to minimize the number of user queries.

**Weaknesses:**

1. The authors acknowledge the limitation of using synthetic data and assuming static user preferences, suggesting future work to address this using real-world data and models of evolving preferences. Still, I find it significantly limits the generalizability of the results. Particularly, the authors propose an extremely complicated algorithm, yet only demonstrate it on synthetic simple environments. While the motivation of their method is clear, the complexity of their approach calls for much more involved experiments, to test the scalability of PASTEL on larger and more complex RL environments with higher dimensional feature spaces. I would also strongly recommend adding user studies with actual human participants to validate the effectiveness and user experience of PASTEL.

2. Real user preferences are likely far more complex, potentially involving non-linear relationships, inconsistencies, and individual variations that are not captured by the simple linear utility model employed with synthetic data. The authors should incorporate user studies with real human participants, evaluating the algorithm's ability to generalize to their utility functions.

3. The experiments use a relatively small and pre-defined feature set for representing DT policies. A broader or automatically discovered feature set could better capture user preferences.

4. The use of an RL agent for initializing the DT policy population has a potential to introduce bias in the initial policy. This initialization could steer the search process toward a region in the policy space that neglects user preferences in favor of higher reward. Although diversity is introduced through modifications to VIPER, the underlying influence of the reward-based initialization could limit the exploration of policies that maximize user utility.

**Questions:**

See above

---

> ### Author Response · Authors · 2024-12-02
>
> Thank you for your thoughtful review! We appreciate that you found the paper to be clear and the algorithm framework to be well designed. We’ll address your comments below:
>
> > The authors acknowledge the limitation of using synthetic data and assuming static user preferences, suggesting future work to address this using real-world data and models of evolving preferences. Still, I find it significantly limits the generalizability of the results. Particularly, the authors propose an extremely complicated algorithm, yet only demonstrate it on synthetic simple environments. While the motivation of their method is clear, the complexity of their approach calls for much more involved experiments, to test the scalability of PASTEL on larger and more complex RL environments with higher dimensional feature spaces. I would also strongly recommend adding user studies with actual human participants to validate the effectiveness and user experience of PASTEL.
>
> Thank you for your thoughtful feedback about generalizability. While we chose synthetic data to establish clear ground truth for preferences, we recognize this creates limitations. The synthetic environments allow us to test PASTEL's various components. However, we agree that testing on more complex environments and with real users would provide valuable additional validation. We appreciate this perspective and will consider how to address these limitations in future work.
>
> > Real user preferences are likely far more complex, potentially involving non-linear relationships, inconsistencies, and individual variations that are not captured by the simple linear utility model employed with synthetic data. The authors should incorporate user studies with real human participants, evaluating the algorithm's ability to generalize to their utility functions.
>
> While non-linear preferences and inconsistencies can certainly exist, the effectiveness of linear utility models in many real-world recommendation systems [1, 2] suggests they can sometimes capture meaningful preference structures. Our synthetic framework allowed us to validate PASTEL's core mechanisms with known ground truth, which was the primary focus of this study. We will consider how to incorporate various functional forms of preferences in future versions of the work.
>
> > The experiments use a relatively small and pre-defined feature set for representing DT policies. A broader or automatically discovered feature set could better capture user preferences.
>
> We would like to clarify that our approach is feature-agnostic: PASTEL can work with any feature generation method, whether pre-defined or automatically discovered. The key is that our framework optimizes over whatever features are provided, without making assumptions about what they represent. One could even leverage language models to generate potential features [3], though that would need validation in these new settings.
>
> > The use of an RL agent for initializing the DT policy population has a potential to introduce bias in the initial policy.
>
> We agree that leveraging a pretrained RL agent could bias the initial policy space. We intentionally center the policy space around reward-relevant regions, based on the practical assumption that users typically have some preference for task success, even if the exact tradeoffs are unknown. This initialization approach helps guide exploration toward policies that balance reward optimization with other user preferences, rather than exploring the entire policy space (which could include many impractical solutions).
>
> To clarify: if maximizing reward isn't a relevant feature for a particular application, we can easily modify the initialization process. The VIPER initialization is a design choice, not a requirement of our framework. Our goal is to start with sensible policies and then refine them based on user preferences, rather than exploring policies that might completely disregard task performance.
>
> ---------------------------------
>
> [1] Jin, Ruoming, et al. "Towards a better understanding of linear models for recommendation." Proceedings of the 27th ACM SIGKDD Conference on Knowledge Discovery & Data Mining. 2021.
>
> [2] Li, Lihong, et al. "A contextual-bandit approach to personalized news article recommendation." Proceedings of the 19th International Conference on World Wide Web. 2010.
>
> [3] Jeong, Daniel P., Zachary C. Lipton, and Pradeep Ravikumar. "Llm-select: Feature selection with large language models." arXiv preprint arXiv:2407.02694 (2024).

---

### Official Review · Reviewer_mqMK · 2024-11-02

**Soundness:** 2
**Presentation:** 1
**Contribution:** 2
**Rating:** 3
**Confidence:** 4

**Summary:**

This paper presents an attempt to incorporate human preferences into the training of interpretable decision-tree (DT) policies for reinforcement learning (RL) using an evolutionary strategy.
They present PASTEL (Preference Aligned Selection of Trees via Evolutionary Learning), a method that employs an evolutionary algorithm to optimise tree policies using feedback from users.
Through iterative comparisons, it adapts policies to improve interpretability and alignment with user preferences, while maintaining good performance on the task at end. The paper includes an experimental evaluation in two RL environments, CartPole-v1 and PotholeWorld-v0, demonstrating that PASTEL produces policies that are preference-aligned and interpretable with high sample efficiency.

**Strengths:**

- The paper and writing are fairly clear
- I thought the Preference Elicitation and Bayesian Experimental Design related literature section was comprehensive and well done
- The authors discuss the issue of balancing optimising human preferences but also succeed at the task at hand, and design an algorithm that takes this into consideration, generating (via the evolution algorithm) diverse methods of solving the task so to get feedback on the preferred method from humans.
- The evolutionary algorithm and the use of Pareto frontier filtering to reduce the number of candidate policies are clever strategies that improve the sample efficiency of the preference elicitation process. This approach minimises the burden on users while still aiming to produce high-quality, preference-aligned policies with limited data points.
- While the use of evolutionary strategies is not groundbreaking, it is nonetheless an interesting application for generating interpretable policies that can adapt based on user feedback. The method effectively addresses the non-differentiability of decision trees, showcasing a practical approach for optimising such structures in an RL setting.

**Weaknesses:**

I believe the work has significant issues in terms of positioning within the broader literature

- The paper fails to connect with key frameworks and research in preference optimization, especially in the context of reinforcement learning from human feedback and large language models. The authors mention that they use preferences during training to align policies with human feedback, but they omit the well-known fact that leveraging preferences is common practice in LLM fine-tuning. Even though this work focuses on interpretable policies rather than neural networks, this comparison is important for understanding the relevance of their suggested approach. Given the work’s emphasis on preference alignment, it is surprising that methods from preference optimization and alignment strategies in LLMs are not discussed at all. The omission creates a significant gap in situating the proposed method in the current research landscape.
- Unclear Scalability and Limited Experiments: The choice of evolutionary strategies to optimise decision trees is interesting but not particularly novel. The experimental validation is also highly limited, featuring only CartPole and PotholeWorld. This narrow scope raises serious questions about the method’s scalability and effectiveness on more complex, real-world problems, and this was not even mentioned in the limitations section. I think a real-world application example where DTs could be effectively used to solve the issue is an important missing point.
- Mention of Imitation Learning / Inverse Reinforcement Learning: I think this is less important than the other weaknesses I listed above, but it is interesting the authors use VIPER (based on DAgger) and Imitation Learning / Inverse Reinforcement Learning are not discussed at all in the background or related work. I think it would have also been interesting to compare or have a comment on preference elicitation vs recovering rewards using Inverse Reinforcement Learning. But this is a nit.
- A nit but I didn't really think the 2 papers referenced on lines 101-102 were particularly related to what was being discussed in that sentence, or were particularly good examples.

**Questions:**

- How does this method scale to more complex environments? And non-synthetic human feedback?
- Why an evolutionary algorithm? Why this choice and what else was considered?
- Decision trees have known limitation in complex / high dimensional data settings, would this apply to complex user feedback too?

---

> ### Author Response · Authors · 2024-12-02
>
> Thank you for your thoughtful review! We appreciate that you found the approach to be practical for adapting interpretable structures based on user feedback and the strategies to improve sample efficiency to be clever. We’ll address your comments below:
>
> >The paper fails to connect with key frameworks and research in preference optimization, especially in the context of reinforcement learning from human feedback and large language models.
>
> We had initially thought that including a lengthy discussion section on this might be confusing, as it could lead a reader to believe that the techniques and motivation are similar. However, as you point out, more explicitly contrasting with this body of work actually can improve clarity. We are happy to update the draft to include a discussion of LLM fine-tuning. In particular, we will highlight that these methods do not immediately give us preferences over attributes of the policies themselves, which is necessary for our setting.
>
> >This narrow scope raises serious questions about the method’s scalability and effectiveness on more complex, real-world problems, and this was not even mentioned in the limitations section. I think a real-world application example where DTs could be effectively used to solve the issue is an important missing point.
>
> We would like to clarify that our work’s novelty is in learning an interpretable policy that adheres to the user’s preference, both the way we do it and the goal as a whole. Using DTs as the policy representation is common in XRL literature [1-3]. We agree they have limitations, and we note that our preference learning elements do not rely on the policy’s form.
>
> That said, we agree that instantiating at least one concrete domain and use case would strengthen the paper. One practical use case is inventory management [4]. As a simple example, an inventory manager might prefer trees where the order of splits more closely aligns with their own mental model of inventory management priorities. Given two trees with similar performance, they would then prefer the policy that more closely aligns with human decision-making.
>
> >Mention of Imitation Learning / Inverse Reinforcement Learning...  I think it would have also been interesting to compare or have a comment on preference elicitation vs recovering rewards using Inverse Reinforcement Learning.
>
> Yes, this is also something that we had considered as a baseline! The challenge here is as follows. Suppose we leverage VIPER. Then, we can use the estimated preference model to update Q to guide VIPER training. If we have preferences over transitions, then this is easy to do, but if we have preferences over policy attributes, then having this Q would not enable us to e.g., obtain a tree of depth 2 that still achieves 80\% optimal reward.
>
> To your point, we can essentially view our framework from the perspective of recovering a preference model to learn a policy (similar-ish to inverse RL). We are happy to add a discussion comparing these approaches to clarify the positioning of the work.
>
> >Why an evolutionary algorithm?
>
> We use an evolutionary algorithm due to the non-differentiable structure of decision trees. We considered relaxing the tree structure and using gradient-based approaches, but the discretization step to improve interpretability often incurs a large performance decrease [1].
>
> >Decision trees have known limitation in complex / high dimensional data settings, would this apply to complex user feedback too?
>
> To clarify, the decision tree doesn’t model the user preference itself. Instead, the policy is represented as a tree. For the user preference model, we assume linearity in features that are functions of a policy decision tree; however, if the preferences are complex we can use a polynomial basis for the features. Exploring what types of complex preferences users could have in an even more practical setting would be an interesting future direction for an HCI study.
>
> -----
> [1] Silva, Andrew, et al. "Optimization methods for interpretable differentiable decision trees applied to reinforcement learning." International conference on artificial intelligence and statistics. PMLR, 2020.
>
> [2] Koirala, Prajwal, and Cody Fleming. "Solving offline reinforcement learning with decision tree regression." 8th Annual Conference on Robot Learning. 2024.
>
> [3] Roth, Aaron M., et al. "Conservative q-improvement: Reinforcement learning for an interpretable decision-tree policy." arXiv preprint arXiv:1907.01180 (2019).
>
> [4] Madeka, Dhruv, et al. "Deep inventory management." arXiv preprint arXiv:2210.03137 (2022).

---

### Official Review · Reviewer_eaLY · 2024-11-03

**Soundness:** 2
**Presentation:** 2
**Contribution:** 1
**Rating:** 3
**Confidence:** 3

**Summary:**

Policies that are decision trees are convenient because they are interpretable. People may also have preferences between such policies, and one may want to elicit preferences from users while learning policies. This work identifies this combination as a novel problem and presents one way to incorporate human preferences as part of the policy training process to favor more interpretable policies parameterized by decision trees.

This paper is essentially doing active learning for user preferences (with a fixed budget for querying users) on decision trees, and also learning decision trees according to these preferences. Each component of the proposed method (how to choose which examples to query user preferences for, how to fit trees to preferences, how to model user preferences) is some sort of heuristic, some of which have been used previously in other settings. The authors demonstrate their method on two synthetic settings, with synthetic preferences.

**Strengths:**

I had not seen this problem setting before, so it seems original to me. The formulation of the setting is written in a clear way. In terms of clarity, the proposed method is explained in the text in a way that is reasonably clear.

**Weaknesses:**

* **Motivation** Although it seems reasonable to want to have preferences over decision trees policies, I am not convinced that the framing as an active learning problem setting is important and realistic (a strong concrete motivating example is not present), and the synthetic nature of the experiments do not help. Also, see questions. The work seems inspired by RL with human feedback, and in that literature, methods are typically developed using preference datasets that are collected offline in a batch, rather than online.

* **Method is entirely heuristic (and evaluations are not realistic)** The proposed method is entirely heuristic and a lot of choices are not really justified (see questions section). This, combined with limited experiments, makes it difficult to understand under what settings it could perform well vs not. Also see questions on hyperparameter choice. I think this would be fine if either the problem setting and experiments were very convincing, or if the method is principled, or you can prove its performance.

* **Experiments are not realistic** The experiments assume that a user's utility is linear in {reward, depth, num leaves, state feature used}, which seems unrealistic. Is there a realistic setting where someone would want a linear tradeoff between reward vs e.g. depth, rather than e.g. put a limit on depth and then maximize reward (which you could do very simply)?

* **Presentation** Although the method is reasonably clear from reading the text, there are still many issues with presentation: the notation is inconsistent, and I found Figure 1 difficult to understand, even though it is visually pleasing.
    * Notation in (3) is inconsistent: sup vs subscripts for $i,j$
    * Notation inconsistency for (4); the text suggests it should be $\hat\Pi_T$ not $\hat\Pi$
    * Indexing for $\hat\theta=\hat\theta_i,\ldots,\hat\theta_N$: what is $i$?
    * Line 213: preference estimates $\Theta=\theta_1,\ldots,\theta_N$. Why the sudden change from $\hat\theta$ (line 194-195) to $\Theta$?
    * Line 200-201: $\hat\theta_t$ where $t$ indexes timestep, not which estimate within a set (ensemble?) of estimates
    * Even more inconsistent notation with line 1 in Algorithm 1
    * What is $m$ in (7)? It does not seem to be the mapping $m$ from (1).
    * Figure 1 would benefit from the thetas being labeled as preference estimates, and the blue stuff being labeled as policy estimates (or similar). I'm not sure what "feature vectors" in Figure 1 refers to. It also has way too many arrows for me to know how I should be reading it. It only made sense after reading the rest of the paper. I don't understand what the grey box is in relation to the rest of the diagram.

**Questions:**

* **Problem setting motivation** What is a concrete and realistic motivating example for this problem setting? Why does the data collection about human policy preferences need to be done at the same time as training? When would this be useful and also realistic?

* **Hyperparameters** How were the hyperparameters (for each of the heuristic components) chosen in the experiments, and how they would be chosen in a real-world setting?

* **Heuristic choice**
    * Why is this novel selection process a good idea? Why does it use the updated $\theta_j^{(i)}$ vs $\theta_j^{(k)}$ rather than the gradient updates $\eta\nabla \mathcal L$ instead? Why is this preference update a good idea? Are there reasons besides that it seems kind of reasonable and seems to perform ok?
    * Why is pareto frontier filtering a good idea besides that it seems reasonable? Are there better alternatives?
    * In practice, how did you choose the hyperparameter $c$ in (9)? Why is a UCB-inspired bound a good idea, besides that it seems reasonable?
    * In what settings would which parts of this method work vs not?

---

> ### Author Response · Authors · 2024-12-02
>
> Thank you for your detailed review! We appreciate that you found the problem setting to be original and clear. We’ll address your comments below:
>
> >Motivation Although it seems reasonable to want to have preferences over decision trees policies, I am not convinced that the framing as an active learning problem setting is important and realistic (a strong concrete motivating example is not present), and the synthetic nature of the experiments do not help. Also, see questions. The work seems inspired by RL with human feedback, and in that literature, methods are typically developed using preference datasets that are collected offline in a batch, rather than online.
>
> We agree that methods for RL with human feedback indeed typically leverage offline preference elicitation methods for developing datasets that capture general human preferences. In contrast, we envision that users may have very specific, in-context use cases that would necessitate online and few queries. In this case, it would be impractical to query a user so many times for their preferences. We appreciate the opportunity to clarify how our motivation differs from the more standard RLHF setup, and we will revise the next draft with a more detailed comparison.
>
> >Method is entirely heuristic (and evaluations are not realistic) The proposed method is entirely heuristic and a lot of choices are not really justified (see questions section). This, combined with limited experiments, makes it difficult to understand under what settings it could perform well vs not. Also see questions on hyperparameter choice. I think this would be fine if either the problem setting and experiments were very convincing, or if the method is principled, or you can prove its performance.
>
> Thank you for the feedback. We simplified the algorithm significantly to begin studying it theoretically. Specifically, we assume 1) trees are compared using random selection, 2) we maintain only a single preference estimate (rather than an ensemble), 3) we leverage the more standard PL model without the temperature parameter $\beta$, and 4) we only run the EA at the end of the $T$ comparisons to obtain the tree that maximizes $\hat{\theta}$.
>
> In this case, we can use the result that the feedback can be modeled as a GLM feedback model on the decision space of pairwise differences [2], meaning that the results from [3] directly apply under suitable conditions. If we obtain sufficient samples to make our estimation error small, then if we assume that the EA can return the tree that is a maximizer, we will obtain a near-optimal solution.
>
> Of note: different from these two prior works, we only care about the simple regret at time $T$, whereas their algorithms are designed to minimize cumulative regret, so their algorithms are not directly applicable.
>
> >Experiments are not realistic The experiments assume that a user's utility is linear in {reward, depth, num leaves, state feature used}, which seems unrealistic. Is there a realistic setting where someone would want a linear tradeoff between reward vs e.g. depth, rather than e.g. put a limit on depth and then maximize reward (which you could do very simply)?
>
> Although linear utility functions are a simplifying assumption, we believe they serve an important purpose. First, linear utility functions are well-established in the contextual bandit literature [2, 4, 5], from which we derive some of our framework. Second, although the function itself is linear, we could transform the feature vector by applying any arbitrary function to the raw input, such as a polynomial basis. Third, the suggestion of using hard constraints (like depth limits) is possible. However, selecting a poor value for the constraint leads to a worse outcome. For example, if a greater depth than necessary is permitted, then the policy will still grow to this greater depth. Likewise, if the depth is set too low, then the policy will perform poorly, and since a deeper policy is not learned, the degree of performance loss will be unknown. Our approach can be seen as learning to select the proper constraints. Fourth, we use the tradeoff between two features for exposition, but we envision even more attributes that users may have preferences over (e.g., the state features tested within the tree policy itself).
>
> >Presentation Although the method is reasonably clear from reading the text, there are still many issues with presentation: the notation is inconsistent, and I found Figure 1 difficult to understand, even though it is visually pleasing.
>
> We really appreciate these detailed suggestions! We will update the paper accordingly.
>
> (1/n)

---

> > ### Author Response · Authors · 2024-12-02
> >
> > >Problem setting motivation What is a concrete and realistic motivating example for this problem setting? Why does the data collection about human policy preferences need to be done at the same time as training? When would this be useful and also realistic?
> >
> > Controllers for industrial cooling systems already use RL [6] and “smart” AC systems for homes already adapt their behavior based on individual user responses. If a user is to interact with a policy (e.g., tweak parameters, seek to understand why a change occurred), then they would have preferences not only about the policy’s outcomes (energy used, state reached) but also the form of the policy (how many nodes is too many? can the policy depend on the day of the week?).
> > By gathering preferences while training, policy space can be searched more effectively. Effectively, a narrower problem is solved: optimizing for a given preference, rather than preemptively learning how to produce a policy to cater to any possible preference.
> >
> > >Hyperparameters How were the hyperparameters (for each of the heuristic components) chosen in the experiments, and how they would be chosen in a real-world setting?
> >
> > Great question. For the EA, we chose large values for the mutation rate and slightly lower values for crossover to prioritize exploration. In a practical setting, choosing the right approximate value could depend on e.g., how likely we think we are to be in a promising region. If we have less confidence (so would benefit from more exploration), we can choose a higher value for the mutation rate. For the fitness evaluation, the user could pre-specify the desired confidence interval.
> >
> > >Why is this novel selection process a good idea?
> >
> > Thanks for the question! The gradient similarity approach doesn't work because for any pair of items $(i,k)$, the two gradients $\nabla \mathcal{L}(f_{\pi_i} - f_{\pi_k}) | \theta_j)$ and $\nabla \mathcal{L}(f_{\pi_k} - f_{\pi_i}) | \theta_j)$ are always exact opposites of each other. This is due to the symmetry of the logistic loss. This means comparing these gradients gives us no information to distinguish between different pairs of items. But perhaps exploring something like gradient magnitudes could lead to a more efficient update.
> >
> > >Why is pareto frontier filtering a good idea besides that it seems reasonable? Are there better alternatives?
> >
> > The motivation of Pareto frontier processing is as follows. In many cases, it is reasonable to assume that we know the direction but not the exact value of a preference. To illustrate with an extremely simple case, we know that people prefer higher reward, so given a set of $k$ policies, we can use this information to choose the policy with the highest reward. But this assumption also extends to other attributes: people generally prefer decreased complexity of explanations. In the context of this framework, filtering according to the Pareto frontier enables us to query only for points that might actually be the maximizer of the user’s preference, by construction of the frontier. One relaxation to explore is simply the convex hull, which leverages no prior information about preference direction.
> >
> > >UCB-inspired bound
> >
> > A UCB-inspired bound enables us to obtain high confidence about our reward estimate with a confidence level that could be pre-specified by a user.
> >
> > >In what settings would which parts of this method work vs not?
> >
> > This is an interesting question that could be explored in future work. For example, we could explore whether the linearity assumption still enables us to obtain reasonable policies under functional form misspecification. Thanks for the suggestion!
> >
> > ----
> >
> >
> > [1] Hunter, David R. "MM algorithms for generalized Bradley-Terry models." The annals of statistics 32.1 (2004): 384-406.
> >
> > [2] Saha, Aadirupa. "Optimal algorithms for stochastic contextual preference bandits." Advances in Neural Information Processing Systems 34 (2021): 30050-30062.
> >
> > [3] Li, Lihong, Yu Lu, and Dengyong Zhou. "Provably optimal algorithms for generalized linear contextual bandits." International Conference on Machine Learning. PMLR, 2017.
> >
> > [4] Li, Lihong, et al. "A contextual-bandit approach to personalized news article recommendation." Proceedings of the 19th International Conference on World Wide Web. 2010.
> >
> > [5] Chu, Wei, et al. "Contextual bandits with linear payoff functions." Proceedings of the Fourteenth International Conference on Artificial Intelligence and Statistics. JMLR Workshop and Conference Proceedings, 2011.
> >
> > [6] Luo, Jerry, et al. "Controlling commercial cooling systems using reinforcement learning." arXiv preprint arXiv:2211.07357(2022).
> >
> > n/n

---

### Official Review · Reviewer_HCiq · 2024-11-08

**Soundness:** 2
**Presentation:** 2
**Contribution:** 1
**Rating:** 5
**Confidence:** 2

**Summary:**

In this paper, the authors propose a framework that can construct decision-tree policies based on human preferences, i.e. paired comparisons. This framework first generates a pool of candidate tree policies using an imitation learning algorithm and then iteratively refine its estimate of the optimal one using an evolution strategy. During the iterative refinement, humans are required to compared policies, which is used to optimize a weighting vector of criteria for policies. The authors validate their method on CartPole and PotholeWorld, showing that their method generates policies that are better aligned with synthetic preferences.

**Strengths:**

1. The proposed framework is clearly presented in the paper.
2. The motivation is clear and persuasive.

**Weaknesses:**

The main weakness of this paper is the set of assumptions used for simplifying the learning problem, which I think are unrealistic. In specific:

1. It is assumed that humans can compare two tree policies and give reliable feedback. While I agree that, tree policies are interpretable in the sense that decision rules can be read from visualizations, it is questionable if humans can compare two trees reliably. For example, this requires humans to precisely predict the consequences of any change in single or multiple leafs. On the contrary, in recent literature of PbRL, humans are only require to compare trajectories, which is much less demanding than comparing policies directly.

2. It is assumed that a d-dimensional feature vector $f_\pi$ can not only summarize the structure of a tree but also all information required for decision-making. This assumption significantly restricts the applicability of this method.

In addition, though interpretability is considered as the motivation of this paper, there is no qualitative results on the extent of interpretability of learned policies.

**Questions:**

1. Can this method be applied to more complex domains, such as the Mujoco tasks (Ant, HalfCheetah, or Humanoid)?
2. Could you provide some examples of preference queries (for both simple and complex tasks) and learned policies as evidence for applicability of this method?

---

> ### Author Response · Authors · 2024-12-02
>
> Thank you for your review! We appreciate that you found the framework clear and the motivation persuasive. We’ll address your comments below:
>
> >It is assumed that humans can compare two tree policies and give reliable feedback. While I agree that, tree policies are interpretable in the sense that decision rules can be read from visualizations, it is questionable if humans can compare two trees reliably. For example, this requires humans to precisely predict the consequences of any change in single or multiple leafs. On the contrary, in recent literature of PbRL, humans are only require to compare trajectories, which is much less demanding than comparing policies directly.
>
> We agree that these UX/UI concerns are important. We envision that users could either be presented with two trees or only the entries captured by our $d$-dimensional feature vector. The latter option circumvents the concern that users may not be able to detect small changes in the tree, as the downstream impact of this change would be captured in a more succinct manner.
> Furthermore, we would like to highlight that we assume a Bradley-Terry preference model that already captures the concern of reliability. Specifically, the Bradley-Terry preference model introduces additional preference noise when the items being compared are closer together. This is a common assumption even in recent work in preference learning literature [1, 2]. Any specific suggestions for more realistic theoretical assumptions underpinning the stochastic feedback model would be appreciated.
>
> >It is assumed that a d-dimensional feature vector  can not only summarize the structure of a tree but also all information required for decision-making. This assumption significantly restricts the applicability of this method.
>
> We would like to emphasize that we are the first to study the problem of aligning interpretable structures with human feedback. As a result, previous work has only assumed that users care about reward under depth constraints [3] or performed human evaluations of interpretability without incorporating this feedback back into the interpretable structure [4]. Both directions neglect capturing potentially complex preference trade-offs that real users may have over various attributes.
>
> We would also like to mention that the assumption that there exists a set of shared attributes that describe sets of items is standard and adopted from literature on contextual bandits for potentially infinite decision spaces [5].​​ The benefit of this approach is that converging to the best item in a set of size $K$ traditionally requires the explicit maintenance of a $K^2$ matrix, which is cumbersome and unnecessary if there is some structured context that is shared among all items.
>
> That said, we appreciate the comment and will consider how to demonstrate the potential cost of misspecification in future work.
>
> >In addition, though interpretability is considered as the motivation of this paper, there is no qualitative results on the extent of interpretability of learned policies.
>
> Thanks for the suggestion! We are happy to provide tree visualizations in future drafts.
>
> >Can this method be applied to more complex domains, such as the Mujoco tasks (Ant, HalfCheetah, or Humanoid)?
>
> Yes, this method could be applied to more complex domains. To handle the continuous action spaces of these domains, the policy trees would be regression trees (rather than classification trees). The other elements are compatible. These environments naturally admit safety-based preferences & trade-offs that could be directly incorporated into our method (e.g., joints should not bend more than X amount to avoid damaging the robot).
>
> (1/2)

---

> > ### Author Response · Authors · 2024-12-02
> >
> > >Could you provide some examples of preference queries (for both simple and complex tasks) and learned policies as evidence for applicability of this method?
> >
> > No problem.
> > 	Users could receive preference queries in one of two forms: 1) trees plus additional information, and 2) only the information provided in the feature vector. We include an example of 2) here for convenience.
> > As a simple example, the user could receive the following information:
> >
> > Tree 1	|		Tree 2
> > -----|----
> > Reward: 36 |	Reward: 34
> > Depth: 10	|	Depth: 4
> >
> >
> > A more complex setting could show:
> > Tree 1			|	Tree 2
> > ----|----
> > Reward: 36		|	Reward: 34
> > Depth: 10		|	Depth: 4
> > Number of leaves: 158 |	Number of leaves: 10
> > Takes action 2?: Yes	| 	Takes action 2?: Yes
> >
> > In the second example, we can see that a user with a stronger preference for reward may be willing to overlook the more complex Tree 1 for a few additional performance points. However, a user that is more sensitive to complexity may choose Tree 2 due to its smaller depth and fewer leaves. In this example, the user actually placed all preference weight on reward, so PASTEL eventually converged on Tree 1 as the best.
> >
> > ----
> > [1] Bai, Yuntao, et al. "Training a helpful and harmless assistant with reinforcement learning from human feedback." arXiv preprint arXiv:2204.05862 (2022).
> >
> > [2] Rafailov, Rafael, et al. "Direct preference optimization: Your language model is secretly a reward model." Advances in Neural Information Processing Systems 36 (2024).
> >
> > [3] Bastani, Osbert, Yewen Pu, and Armando Solar-Lezama. "Verifiable reinforcement learning via policy extraction." Advances in neural information processing systems 31 (2018).
> >
> > [4] Boyd-Graber, Jordan, et al. "Human-centered evaluation of explanations." Proceedings of the 2022 Conference of the North American Chapter of the Association for Computational Linguistics: Human Language Technologies: Tutorial Abstracts. 2022.
> >
> > [5] Saha, Aadirupa. "Optimal algorithms for stochastic contextual preference bandits." Advances in Neural Information Processing Systems 34 (2021): 30050-30062.
> >
> > (2/2)

---

### Meta-Review · Area_Chair_cKtr · 2024-12-20

**Metareview:**

The paper proposes a preference learning technique for aligning decision trees with human feedback. All of the reviewers agreed that although the paper clearly explains its problem framing and solution approach, it lacks in key areas like motivation and execution.

The following will substantially strengthen the paper:
1. A user study to verify that humans are indeed able to provide feedback contrasting two decision trees in the quantity and quality needed for the proposed method.
2. Motivate why the data collection about human policy preferences needs to be done at the same time as training the decision trees.
3. Run the proposed technique on several more domains, run ablations to identify what components contribute the most towards the overall performance.
4. Discuss non-linear and other forms of human preferences (e.g. constrained, multi-objective, etc.) and how the linear utility model assumed in the method may perform under such mis-specification.

**Additional Comments On Reviewer Discussion:**

The authors clarified their core contributions and addressed some of the conceptual questions raised by the reviewers. They acknowledged the connections in the RLHF, LLM alignment and inverse RL literatures. However the substantive weaknesses mentioned above (lack of user study, motivation, ablation experiments, and preference modeling variants) remain.

---

### Decision · Program_Chairs · 2025-01-22

Reject